# US Nicotine Vaping Product *SimSmoke* Simulation Model: The Effect of Vaping and Tobacco Control Policies on Smoking Prevalence and Smoking-Attributable Deaths

**DOI:** 10.3390/ijerph18094876

**Published:** 2021-05-03

**Authors:** David T. Levy, Luz María Sánchez-Romero, Nargiz Travis, Zhe Yuan, Yameng Li, Sarah Skolnick, Jihyoun Jeon, Jamie Tam, Rafael Meza

**Affiliations:** 1Lombardi Comprehensive Cancer Center, Georgetown University, Washington, DC 20007, USA; ls1364@georgetown.edu (L.M.S.-R.); nt526@georgetown.edu (N.T.); zy70@georgetown.edu (Z.Y.); yl954@georgetown.edu (Y.L.); 2Department of Epidemiology, University of Michigan, Ann Arbor, MI 48109, USA; srskol@umich.edu (S.S.); jihjeon@umich.edu (J.J.); rmeza@umich.edu (R.M.); 3Department of Health Policy and Management, Yale University School of Public Health, Hartford, CT 06520, USA; jamie.tam@yale.edu

**Keywords:** smoking, cigarettes, e-cigarettes, ENDS, vaping, tobacco control, simulation model, public health

## Abstract

The public health impact of nicotine vaping products (NVPs) is subject to a complex set of uncertain transitions between NVP and cigarette use. Instead, we apply an indirect method to gauge the impact of NVP use on smoking prevalence and smoking-attributable deaths (SADs) using the well-established *SimSmoke* tobacco control policy simulation model. Upon validating the model before NVPs were more widely used, we project a No-NVP (i.e., in the absence of NVPs) while controlling for the impact of cigarette-oriented policies. The net impact of NVPs on smoking prevalence is inferred by comparing the projected No-NVP smoking trends to corresponding trends from two US national surveys. Using the TUS-CPS estimates for the period 2012–2018, we estimate that adult smoking prevalence declined in relative terms by 9.7% (95% CI: 7.5–11.7%) for males and 10.7% (95% CI: 9.1–13.0%) for females. Compared to NHIS, smoking prevalence declined by 10.7% (95% CI: 6.8–14.6%) for males and 11.3% (95% CI: 7.4–15.6%) for females. These impacts were confined mainly to ages 18–44. Vaping-related reductions in smoking prevalence were projected to avert nearly 0.4 million SADs between 2012 and 2052. Our analysis indicates that NVP use is associated with substantial reductions in US smoking prevalence among younger adults.

## 1. Introduction

Since the Surgeon General’s Report in 1964, smoking prevalence has declined by more than 50% in the US [1,2]. Simulation models show that much of this reduction can be explained by smoking tobacco control policies [1,2,3,4,5], especially increased cigarette taxes, smoke-free air policies, media campaigns, restrictions on youth access to tobacco, and cessation treatment policies [1,6]. While simulation models project that US smoking prevalence will continue to decline under current policies [3,7,8,9,10,11], they also indicate that prevalence will not drop below 10% for at least another two decades. That trajectory may change now that NVPs are more widely used. As a country with relatively unrestrictive policies [12,13], the US provides a case study of the potential impact of NVPs on smoking.

NVPs represent a new generation of nicotine delivery products, which have become progressively more efficient in delivering nicotine to the user [14,15,16]. Studies indicate that NVP use has enabled smokers to quit smoking [17,18,19,20,21] and has replaced cigarette smoking among some youth and young adults [22,23,24]. Other studies suggest that NVP use may lead to youth cigarette smoking [25,26], inhibit smoking cessation [27,28] and promote relapse [29,30]. Predicting the impact of NVPs on smoking prevalence depends on the ability to accurately incorporate transitions between NVP use and smoking [31]. However, that data are often limited and, as a disruptive technology [32,33], NVP use and associated transitions may be unstable.

Rather than explicitly modeling the impact of NVPs on smoking prevalence, we apply an indirect method that does not require information on smoking and NVP use transitions. We first develop a No-NVP scenario, which represents smoking behavior assuming the absence of NVPs. To obtain the No-NVP scenario, we adapt the well-established and previously validated US *SimSmoke* tobacco control policy simulation model [3,7,34], which controls for the effect of cigarette-oriented policies independent of NVP use. We then compare these No-NVP projections to actual smoking rates derived from national surveys to estimate the net impact of NVPs on smoking prevalence and smoking-attributable deaths (SADs).

## 2. Methods

We first validate the model over the time period 1993–2012, before NVP use became more widespread. That model is then used to extrapolate trends in smoking prevalence in the absence of NVPs in the post-NVP period of 2012–2018, which is then compared to actual smoking prevalence.

### 2.1. SimSmoke and the No-NVP Counterfactual

The *SimSmoke* consists of separate components for the population by smoking status, smoking-attributable deaths, and tobacco control policies for years 1993 and onward [35,36]. The model is described briefly below and in more depth in a Technical Report available from the authors.

Actual and projected population estimates by age and gender were obtained from the US Census Bureau for years 1993–2052 [37,38]. We incorporated net international migration [39] and death rates distinguished by smoking status [2,40] to the population as it evolved from 1993. The model population projections through 2052 were validated against population estimates (1993–2015) and projections (2016–2052) by the US Census Bureau.

Data from the Tobacco Use Supplement-Current Population Survey (TUS-CPS) [41] were used to initialize current, former and never smoking prevalence in 1993. Current smokers were defined as individuals who have smoked more than 100 cigarettes in their lifetime and currently smoke either daily or some days. Former smokers were defined as those who were smokers but report that they are not smoking and were differentiated by the number of years since quit smoking. Never smokers have not smoked 100 cigarettes in their lifetime.

Transitions are measured net of death rates distinguished by smoking status. The death rates for never and current smokers by age, gender and year from 1993 to 2052 were based on estimates from the Cancer Intervention and Surveillance Modeling Network (CISNET) Lung consortium [2,40,42]. To estimate death rates for former smokers by years quit, the relative risk (RR) of former smokers by years quit was derived by first taking the ratio of current to never smoking death rates, i.e., smoker RR by age, gender and year and then applying a log-linear reduction to smoker RRs based on past studies as follows: [43,44,45,46,47] former smoker RR = exp[X * log(smoker RR)], where X = 100% for quit < 1 year ago, 92.0% for quit 1–2 years, 79.0% for quit 3–5 years, 58.0% for quit 6–10 years, 32.5% for quit 11–15 years, and 8.0% for quit > 15 years. Former smoker death rates by years quit were then obtained by multiplying the current smoker death rate by the ratio of the relative risks of current to former smokers.

A discrete, first-order Markov process was employed to simulate smoking initiation (never smoker to current smoker), cessation (current smoker to former smoker) and relapse (former smoker to current smoker). Because our goal was to develop initiation rates that reflect our measure of current smokers and because cessation is difficult to accurately measure at early ages, we measured net initiation by subtracting the 1993 current smoking prevalence at two consecutive ages divided by the 1993 never smoking prevalence at the younger age to estimate the 1993 initiation rates net of cessation, e.g., initiation at age 16 = (1993 smoking prevalence at age 16 − 1993 smoking prevalence at age 15)/1993 never smoking prevalence at age 15. Net initiation was measured from age 10 until the ages when smoking prevalence peaked: 22 for males and 25 for females. Cessation starts from the last age of net initiation and was obtained from the TUS-CPS. Cessation was measured as those who quit in the last year but not in the last three months [48]. This measure implicitly assumes that those who quit in the last 3 months (and are not counted) are offset by former smokers who quit but relapsed in months 3–12. Age- and gender-specific relapse rates [49,50,51] by years quit were then applied.

*SimSmoke* incorporates the impact of policies on initiation and cessation. The effects of the policies are generally applied as percentage reductions in smoking rates in the year when a new policy is implemented and as percentage change to initiation and cessation rates in future years. The percent reductions are applied multiplicatively when multiple policies are simultaneously in effect. *SimSmoke* policies and policy effect sizes are shown in Table 1. Policy levels are tracked over time beginning with their initial level in 1993. The effect of a newly implemented policy depends on the incremental effect from previous levels of the same policy.

To gauge the tax/price policy effect, price changes are translated into changes in smoking prevalence through an equation dependent on price elasticities which vary by age [3,6,52]. Cigarette prices and taxes through 2019 were obtained from Tax Burden of Tobacco reports [53], and were adjusted for inflation using the Bureau of Labor Statistics (BLS) consumer price index [54].

Smoke-free air laws incorporate restrictions on (1) worksites, (2) restaurants, (3) bars, and (4) other public places [6,55]. The impact of smoke-free air laws depends on the extent of the ban and level of enforcement (based on compliance data) and publicity (based on the extent of media campaigns). Data on smoke-free air laws are from the American Nonsmokers Rights Foundation website by state and locality and is weighted by population [56]. For example, strict bans in 2019 were implemented in 76.1% of US worksites, 77.8% of US restaurants, 66.4% of US pubs and bars, and 95% of US other public places. Enforcement is set at level 8 out of 10 in all years [7].

Mass campaigns are gauged by average per capita expenditures for media campaigns, school education programs, and community interventions [6,57]. Total tobacco prevention spending [58] is divided by the US population to obtain spending per capita. Based on average expenditures, the policy level was set at 90% minimal and 10% moderate level from 1993 to 1994, increasing to 100% moderate level in 2003. The level was reduced back to 50% minimal and 50% moderate level during the period 2011–2017 but returned to a 25% minimal and 75% moderate level in the period 2018–2019 [3,7].

Marketing restrictions correspond to the type of bans on advertising and marketing and depend on enforcement [6]. Federal laws prohibit cigarette advertising, except at retail point-of-sale and in newspapers and magazines. Restrictions on advertising were set to a minimal level from 1993 through 2009, then increased to 25% moderate and 75% minimal level in 2010 with added FDA restrictions. Enforcement was set at level 9 out of 10 for all years [3,7].

US health warnings were maintained at a low level from 1993 to 2019 since text-only warnings covered less than 10% of packages for all years.

Cessation treatment policies include pharmacotherapy (PT) availability, financial coverage of treatments, quitlines and health care provider brief interventions [59,60]. Nicotine replacement therapy (NRT, gum) has been available since 1988, with the addition of the patch in 1993. Bupropion has been provided with a prescription since 1998. Treatment coverage was initiated in stages beginning with 30% coverage in 1997 increasing to 40% in 2002, 50% in 2007, and 75% in 2014. A national quitline was implemented from 50% in 2003 increasing in stages to 90% in 2007. Brief interventions are set at 50% coverage for all years [59].

Youth access policy considers the effect of retail compliance with minimum purchase age restrictions and self-service and vending machine bans [61]. Youth access enforcement increased to low level in 1998 and to mid-level in 2003, based on non-compliance rates of about 10% [3,7].

Vending machine ban coverage increased from 50% in 1993 to 75% by 2000 and 100% by 2010. Self-service bans begin at 50% in 1995 increasing to 100% in 2010 [3,7].

### 2.2. Calibration and Validation

We first validated the model prior to 2012, before NVP use became more widespread in the US. NVP use was minimal in the US until 2013 [23,62,63], when third-generation vaping devices became available that delivered nicotine more efficiently than previous-generation devices [14,15,16]. Adult NVP use increased substantially and reached 3–5% in 2013, with as much as 20% use among current and recent former smokers [62,64,65]. Therefore, we designate the years 1993–2012 as the pre-NVP period, which is used to project the No-NVP counterfactual after 2012.

*SimSmoke* smoking prevalence was first calibrated against prevalence estimates by age and gender from the TUS-CPS and the National Health Interview Survey (NHIS) over the period 1993–1998, a period of little policy change. For 1993 and the ensuing years, the initiation rates were increased by 30% for males and 15% for females primarily to reflect the increasing smoking prevalence among young adults over that time period. The increase in smoking among youth has been associated with the “Joe Camel” campaign in the early nineties [66,67,68].

We then validate over the pre-NVP period. *SimSmoke* smoking prevalence projections by gender and age group are compared against national surveys. With TUS-CPS available every three to four years, we first validated through 2010/11, the year closest to and before 2012. We also use estimates from the annual NHIS to validate the model through 2012. Because initial prevalence levels differ across surveys and from *SimSmoke,* we compared relative changes in smoking prevalence from 1998 to 2012 as [(SmokPrev_2012_–SmokPrev_1998_)/SmokPrev_1998_]. We also examined whether *SimSmoke* projections are within the confidence intervals (CI) of the survey estimates. The validated 1993–2012 model is then used to extrapolate post-2012 trends in smoking prevalence under the No-NVP counterfactual.

### 2.3. The Impact of NVPs

The impact of NVPs is inferred by comparing smoking projections by age and gender in the period 2012–2018 under the No-NVP counterfactual scenario to actual smoking prevalence during the post-NVP 2012–2018 period from surveys. To estimate the changes in smoking prevalence reflective of NVP use, we subtract the *SimSmoke* projected relative change in smoking prevalence between 2012 and 2018 in the No-NVP counterfactual scenario from the corresponding estimated relative change in smoking prevalence from TUS-CPS and NHIS. Because the TUS-CPS is not available in 2012, the prevalence in that year is estimated as a weighted average of TUS-CPS estimates in 2010/2011 (60% weight) and 2014/15 (40% weight). We tested for differences in the counterfactual and actual estimates by considering whether the projected 2012–2018 *SimSmoke* relative changes are outside of the corresponding CIs from the TUS-CPS and NHIS relative changes.

As a gauge of uncertainty for the implied NVP impacts, we use the 95% CIs of the 2018 survey estimates as upper and lower bound estimates for smoking prevalence in 2018. We then estimated net NVP impacts by comparing relative changes in *SimSmoke* projections to the relative reductions based on the upper and lower bounds (e.g., upper bound of the relative reduction = (upper bound of 2018 prevalence-2012 prevalence)/2012 prevalence). We also conducted sensitivity analyses based on the uncertainty around the policy effects in our 2012–2018 projections, by applying upper and lower bounds for each policy’s effects (+/−25% of the effects for taxes and +/−50% of the effects for other policies) based on a literature review [6].

To estimate the impact of NVPs on smoking-attributable deaths (SADs), we first obtained the NVP-adjusted smoking prevalence by incorporating the inferred NVP-related reduction in smoking prevalence for each survey back into the model. Separate yearly adjustments by gender and age are used to obtain the NVP-adjusted *SimSmoke* prevalence, i.e., the No-NVP projections each year were reduced in relative terms by the average annual reduction rate [(1- survey smoking relative reduction in the period 2012–2018)^1/6^—(1- *SimSmoke* smoking relative reduction in the period 2012–2018)^1/6^], which is also called the NVP adjustor. The NVP-adjusted annual reduction is transferred to never smokers if under the age of 25 or former smokers if above 25.

Once the NVP-adjusted smoking rates were obtained, we separately estimate SADs under the No-NVP and NVP-adjusted *SimSmoke* by multiplying the number of current smokers at each age by their excess mortality risks (current smoker death rate—never smoker death rate). The same procedure is applied to former smokers and summed over current and former smokers to obtain total SADs. The potential public health impact of 2012–2018 NVP use is measured as the difference in SADs between the NVP-adjusted and the No-NVP *SimSmoke* between 2012 and 2052.

## 3. Results

### 3.1. Validation of Smoking Prevalence Estimates over the Pre-NVP Period

As shown in Figure 1a,b, the projections for males and females in the No-NVP counterfactual scenario are compared to actual trends, especially for the post-NVP period (after 2012). The gender- and age-specific *SimSmoke,* TUS-CPS and NHIS smoking prevalence estimates from 1993 to 2012 are presented in Table 2. Validating against the survey estimates over the period 1998–2010, *SimSmoke* yielded a relative reduction in adult (age 18+) male prevalence of 28.8% compared to 29.6% from TUS-CPS and 18.6% from NHIS. The *SimSmoke* projected 2010 male smoking prevalence was 17.9% compared to 17.2% from TUS-CPS and 21.5% from NHIS. For adult females, *SimSmoke* yielded a relative reduction of 30.8% compared to 31.3% from TUS-CPS and 21.4% from NHIS. The *SimSmoke* 2010 female smoking prevalence was 14.0% compared to 13.7% from TUS-CPS and 17.3% from NHIS. Between 1998 and 2012, *SimSmoke* adult male prevalence declined by 31.2% (2012 prevalence: 17.3%) compared to 22.3% from NHIS (2012 prevalence: 20.5%). *SimSmoke* female smoking prevalence declined by 33.3% (2012 prevalence:13.5%) compared to 28.2% from NHIS (2012 prevalence:15.8%).

For ages 18–24, *SimSmoke* male smoking prevalence declined by 31.6% (2010 prevalence: 19.8%) from 1998 to 2010 compared to 35.4% from TUS-CPS (2010 prevalence: 19.4%) and 27.2% from NHIS (2010 prevalence: 22.8%). For female counterparts, the relative reduction was 34.4% from *SimSmoke* compared to 40.3% from TUS-CPS and 29.0% from NHIS. For males of ages 25–44, the relative reductions was 28.1% from *SimSmoke* compared to 30.9% from TUS-CPS and 17.3% from NHIS. For females of ages 25–44, the relative reduction from *SimSmoke* was 31.0% compared to 34.2% from TUS-CPS and 22.7% from NHIS. For males of ages 45–64, the *SimSmoke* relative reduction was 26.3% compared to relative reductions of 25.6% from a TUS-CPS and 16.2% from NHIS. For females of ages 45–64, the relative reduction from *SimSmoke* was 28.8% compared to 22.2% from TUS-CPS and 15.1% from NHIS. For males of ages 65 and above, the *SimSmoke* relative reduction was 26.8% compared to 20.1% from TUS-CPS and 6.7% from NHIS. For females of ages 65 and above, the *SimSmoke* relative reduction was 27.0% from compared to 28.6% from TUS-CPS and 17.0% from NHIS.

The levels of smoking prevalence in 2010 and the relative reductions between 1998 and 2010 from *SimSmoke* were generally between those of the NHIS and TUS-CPS but closer to those of the TUS-CPS for the different age groups. All *SimSmoke* estimates of 2010 smoking prevalence were within the 95% CIs for TUS-CPS estimates, except for females of ages 45–64 (within 0.1%) and males 18+ (within 0.3%). There was less alignment with NHIS. Relative reductions from 1993 to 2012 from *SimSmoke* were greater than those from NHIS, except for ages 18–24 (and 65+ for women), but the NHIS relative reductions were sensitive to years considered in the estimation.

### 3.2. Impact of NVPs on Smoking Prevalence Relative to a No-NVP Scenario, 2012–2018

For the period 2012–2018, Table 3 shows the yearly levels and projected relative reductions in smoking prevalence from the *No-NVP SimSmoke* (counterfactual), TUS-CPS and NHIS. Table 3 also shows the net impact of NVPs as the difference between the *No-NVP SimSmoke* and survey relative reductions. The projected smoking prevalence for age 18 and above in the period 2012–2018 and the data from surveys are also illustrated in Figure 1a,b.

For adult males, *No-NVP SimSmoke* projects a relative reduction in smoking prevalence of 12.2%. The TUS-CPS shows a relative reduction of 21.9% (95% CI = 19.7–23.9%), implying a 9.7% (95% CI = 7.5–11.7%) net NVP-related reduction, while NHIS yields a relative reduction in prevalence of 22.9% (95% CI = 19.0–26.8%), implying a 10.7% (95% CI = 6.8–14.6%) net NVP-related reduction. For adult females, net relative reductions of 10.7% (95% CI = 9.1–13.0%) are implied using TUS-CPS and 11.3% (95% CI = 7.4–15.6%) using NHIS. For both genders, the *No-NVP SimSmoke* projected level is outside of the 95% CI levels in 2018 for NHIS and TUS-CPS and the 2012–2018 relative reductions in smoking prevalence are outside of the 95% CI levels of the relative reductions for both TUS-CPS and NHIS.

The largest relative reductions in smoking prevalence occur among younger age groups. For ages 18–24, the net NVP-related relative reductions of 43.2% (95% CI = 36.4–49.3%) for males and 45.9% (95% CI = 38.9–52.4%) for females are implied by TUS-CPS, compared to 52.6% (95% CI = 42.6–63.0%) for males and 44.4% (95% CI = 29.9–58.8%) for females from NHIS. For ages 25–44, TUS-CPS implies 16.8% (95% CI = 13.5–19.8%) male and 19.8% (95% CI = 16.4–22.4%) female net reductions in smoking prevalence, while NHIS implies 17.3% (95% CI = 11.0–23.6%) and 11.1% (95% CI = 3.8–18.4%) reductions for males and females, respectively. For ages 45–64, TUS-CPS has net relative increases of 2.5% (95% CI = −0.8–5.8%) for males and 3.1% (95% CI = 0.0–6.5%) for females, while NHIS shows a net relative increase of 8.1% (95% CI = 1.1–14.9%) for males but a net relative decrease of 6.9% (95% CI = 0.4–13.1%) for females. For ages 65 and above, TUS-CPS shows net relative increases of 8.2% (95% CI = 2.8–14.4%) for males and 0.8% (95% CI = −4.1–6.2%) for females, compared to 5.2% (95% CI = −6.6–16.1%) male and 5.2% (95% CI = −7.0–17.0%) female net relative increases from NHIS.

For the 18–24 and 25–44 age groups, the *No-NVP SimSmoke* projected overall and the converted annual relative reduction by 2018 are outside the 95% CIs of relative changes from both TUS-CPS and NHIS for both genders. However, for the 45–64 and 65+ age groups, the *No-NVP SimSmoke* projected overall and annual relative changes are generally within the 95% CIs of relative changes from both TUS-CPS and NHIS for both genders. Thus, only those below age 45 show clear differences between the surveys and No-NVP *SimSmoke* projected trend.

### 3.3. Impact of Policies in the Post-NVP Period

Since the estimates of NVP-related impacts on smoking depend on the ability to control for policy changes from 2012 to 2018, we conducted a sensitivity analysis of the impact on *SimSmoke* projections from 2012 to 2018 by applying changes in policies shown in Table 4. The relative reductions in smoking prevalence from 2012 to 2018 increases with the magnitude of policy impact, and consequently the total NVP impact, i.e., their difference from actual smoking prevalence, increases with the magnitude of policy impact. Applying upper and lower bounds for each policy’s effects [6], the relative reduction in adult smoking prevalence with all policies implemented ranges from 10.8% to 13.6% for males and from 11.4% to 14.2% for females, of which 0.34% were attributed to price, 0.34–0.35% to cessation treatment, 0.55–0.60% to smoke-free air laws, and 0.07–0.09% to mass media campaigns. The absolute reduction in *SimSmoke* smoking prevalence projections due to policy changes from 2012 to 2018 is about 1.4%, representing a relatively small impact. Compared to the implied NVP impact on male smoking prevalence from TUS-CPS (9.7%), the variation in impact is one-seventh (i.e., 1.4%/9.7%, where 9.7% is the difference between the *SimSmoke* projection and TUS-CPS estimate).

### 3.4. Long-Term Impact of NVP Use during the Period 2012–2018 on Future Smoking-Attributable Deaths

Table 5 shows the smoking-attributable deaths in total and deaths averted over the time period 2012–2052 based on the implied net NVP-related reductions in the period 2012–2018. Because only the *SimSmoke* projections for those below ages 45 are outside the 95% CIs of prevalence from the surveys, we estimated SADs averted excluding those ages 45 and above. Applying the TUS-CPS adjustment, 273,632 (225,991–317,096) male and 108,319 (91,801–122,168) female SADs are averted, yielding a total of 381,952 (317,792–439,264) averted deaths. Applying the NHIS adjustment, 291,053 (203,156–381,133) male and 74,269 (39,186–110,085) female SADs are averted, a total of 365,322 (242,342–491,218) averted deaths.

As sensitivity analysis, we included deaths for ages 45 and above along with those at younger ages. As a result of findings that NVP use may increase smoking for some older age groups, the number of averted deaths fell slightly to 367,477 (158,716–559,060) using TUS-CPS and 355,549 (−52,783–775,526) using NHIS.

## 4. Discussion

Since stable and reliable estimates of vaping and smoking transitions are difficult to obtain, we applied an indirect method for estimating the effect of NVP use on smoking prevalence and related SADs. Over the period 2012–2018, the implied NVP-related reductions in smoking prevalence were estimated at 10–11% for male and female adults ages 18 and above, with reductions ranging from 43 to 53% for ages 18–24 and 11 to 20% for ages 25–44. The results for young adults are broadly consistent with other modeling studies [69,70], and empirical studies that show higher NVP use in this age group [62,64,65,71,72] and related reductions in smoking prevalence [19,23,26,73,74,75,76,77]. At older ages, the estimates often implied NVP-related increases in smoking (e.g., through reduced cessation or increased relapse [28,29]), but *SimSmoke* estimates were generally within CIs of survey estimates thus suggesting no differences. When confining the implied NVP impact to ages 18–44, the NVP-related reductions increased to 17–20% for males and 12–15% for females.

Our findings for all US adults are similar to those from the *England SimSmoke* model that applied the same methodology [78]. We obtained reductions of 14–20% for males and 13–20% for females from 2012 to 2018. However, unlike the US, the effects for England were more evenly distributed across age groups. England, like the US, has less strict NVP regulations than most other countries. However, England has been regulating NVPs longer and with a broader range of policies (e.g., regarding advertising and product content), and has also begun recommending vaping to help smokers who have not otherwise been successful to quit [79,80]. Additionally, England has less NVP use among youth than the US, which may be due to lower nicotine limits set on products such as JUUL in comparison to the US [81,82,83,84].

Using the US model, we estimated the implied impact of the 2012–2018 NVP-related reductions in smoking prevalence on SADs. For adults ages 18 and above, *SimSmoke* projected close to 0.4 million SADs averted between 2012 and 2052 using both the TUS-CPS-adjusted and NHIS-adjusted NVP impacts. We note that these reductions increase over time, because much of the effect of NVPs is for those at younger ages and deaths attributable to smoking occurring largely at later ages. Some of the reduction in SADs from reduced smoking would be offset by NVP-attributable deaths, but the mortality risks of NVP use, although uncertain, appear to be lower than for smokers [26,80,85]. Our estimates are based on NVP-related reductions in smoking prevalence from 2012 to 2018 but further reductions could occur after 2018.

Our results are subject to limitations. The impact of NVPs is inferred based on a model that controls for past trends in smoking prevalence and the impact of past and newly implemented tobacco control policies but does not explicitly incorporate the impact of NVP use. This method implicitly assumes that other than past and newly cigarette-oriented policies, vaping is the only factor that would have influenced smoking prevalence during the post-vaping period. However, the inferred impact of NVPs may be due to other factors not incorporated into our model, such as changes in attitudes toward cigarette smoking or tobacco control policies not explicitly considered. For example, Food and Drug Administration regulatory efforts [86], newly imposed advertising restrictions in 2018 [87], smoke-free public housing requirements [88] and national media campaigns such as the “Real Cost” campaign [89,90,91] may not have been fully captured by the model.

A second limitation is that the implied NVP impacts rely on the validity of the *SimSmoke* model. *SimSmoke* has been validated for the US [3,7,33] and many countries [92,93,94,95,96,97,98,99,100,101,102,103,104], and the US version of *SimSmoke* applied here was validated against the TUS-CPS and NHIS for the pre-NVP years 1993–2012. Our indirect method also implicitly assumes that the effects of cigarette-oriented policies are the same in the pre- and post-NVP period. However, the effects of policies targeting cigarettes may be altered by the availability of NVPs. While NVPs may blunt the impact of some cigarette-oriented policies (e.g., use of NVPs in areas that are smoke-free, but not vape-free) [105], NVPs may instead enhance policy impacts, because smokers may substitute NVPs for cigarettes in response to stricter cigarette-oriented policies. Indeed, demand studies [106,107,108,109] indicate that NVPs are a substitute for cigarettes, and cessation studies [17,18,19,20,110,111] indicate that NVPs are often used by those who have had limited success with traditional cessation treatments. However, any policy-enhancing effect of cigarette-oriented policies due to NVPs should be included as a benefit of NVP and thus incorporated in our calculations.

Another limitation of the model is that the NVP-attributable impacts depend on the accuracy of survey estimates, which varied considerably in terms of levels and trends between TUS-CPS and NHIS. Further study is needed to explain the variation in smoking prevalence estimates between the NHIS and TUS-CPS.

The projected survey trends may depend on how the post-NVP period is defined. While relative reductions of 10%-11% were observed during the period 2012–2018, shifting to 2011 or 2013 as initial NVP years yielded similar reductions of 12–16% or 11–14%, respectively. We note, however, that model validation was impacted by the pre-NVP period considered.

Finally, our analysis only considered potential impacts of NVPs on smoking. A complete public health assessment would require consideration of the impact of NVP use on health outcomes [30]. If NVPs increase mortality risk, then the increased deaths due to NVP use would reduce the number of total deaths averted. NVP use would improve public health to the extent that they replace smoking by those who would have otherwise initiated cigarette use or who would not have otherwise quit smoking, and negatively impact public health to the extent that it acts as a gateway to smoking or is used as a substitute for smoking cessation. In particular, policies may be needed to reduce NVP use among youth if NVPs act as a gateway to smoking.

## 5. Conclusions

Our model results imply that historical and ongoing tobacco control policies in the US cannot fully explain the accelerated reductions in smoking prevalence in the time period when NVP use became more prevalent. These reductions were found mainly among those at younger ages who are more likely to use NVPs. While our results suggest an important role of NVPs in reducing smoking prevalence, further research is needed to explicitly evaluate net changes in smoking initiation and cessation associated with NVP use, and how these changes could be influenced by future cigarette-oriented and NVP-oriented policies.

## Figures and Tables

**Figure 1 ijerph-18-04876-f001:**
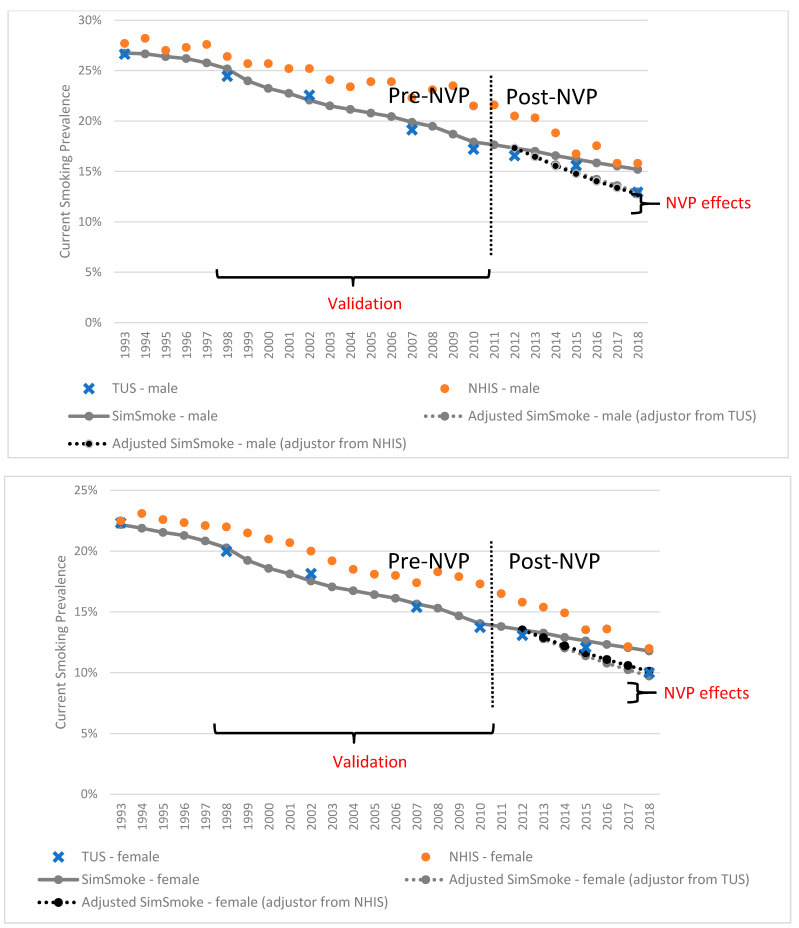
Smoking prevalence in surveys and the SimSmoke model with/without the NVP adjustors by sex, ages 18 and above, 1993–2018.

**Table 1 ijerph-18-04876-t001:** Tobacco control policies, specifications and effect sizes applied in US *SimSmoke*.

Policy	Description				Policy Effect Size		Policy Level, 1993–2019		
**Cigarette Excise Taxes**									
Cigarette price/tax	The effect of taxes is directly incorporated through the average price after tax. The price elasticity is used to convert the price changes (%) into effect sizes				Elasticities		The inflation-adjusted cigarette price increased from $1.75 per pack in 1993 to $3.6 in 2002 to $5.60 in 2012 and $6.60 in 2019.		
					−0.6 for ages 14–17				
					−0.4 for ages 18–24				
					−0.2 for ages 25–34				
					−0.1 for ages 35–64				
					−0.2 for ages 65+				
**Smoke-Free Air Laws**									
Worksite smoking ban	Ban in all indoor worksites, with strong enforcement of laws (reduced by 1/3 if allowed in ventilated areas and by 2/3 if allowed in common areas)				−6% prevalence and initiation, +6% cessation		Worksite ban was at 37% low 7% mid and 1.5% high with little increase through 2002 and gradually increased to 76.1% high and 10.4% mid and 13.5% low in 2019. *Restaurant ban* were less than 1% before 2002 and gradually increased to 77% by 2014 with little further change. *Bars ban* was 0 until 2001 and gradually increased to 65% by 2014. Ban in other places was 50% before 1999 increasing to 94.8% by 2012.		
Restaurant smoking ban	Ban in all indoor restaurants (scaled for lower coverage), with strong enforcement of laws				−2% prevalence and initiation, +2% cessation				
Pubs and bars smoking ban	Ban in all indoor in pubs and bars (scaled for lower coverage), with strong enforcement of laws				−1% prevalence and initiation, +1% cessation				
Other place bans	Ban in 3 out of 4 government buildings (scaled for lower coverage), retail stores, public transportation, and elevators, with strong enforcement of laws				−1% prevalence and initiation, +1% cessation				
Enforcement and Publicity	Government agency enforces the laws and publicity via tobacco control campaigns				Enforcement is ranked on a 1–10 scale converted to percentage terms and publicity is based on indicator = 1 if media campaigns are at a medium level. Effects reduced 50% absent publicity and enforcement; Effect sizes are deflated by: 0.5*(1 + 0.5* Publicity Indicator + 0.5* Enforcement Level).		The enforcement level is 8 out of 10 in all years and the publicity level is based on the level of the media campaigns.		
**Media Campaigns**									
High level media campaign	Campaign publicized heavily with state and local programs with strong funding (>$0.50 USD)				−6.5% prevalence and initiation, +6.5% cessation		Campaigns at 90% minimal and 10% moderate level, increasing to 100% moderate level in 2003, and reduced back to 50% minimal and 50% moderate level from 2011 to 2017, then returning to a 25% minimal and 75% moderate level in the period 2018–2019.		
Medium level media campaign	Campaign publicized with funding of at least $0.10 USD per capita				−3.25% prevalence and initiation, +3.25% cessation				
Low level media campaign	Campaign publicized only sporadically with minimal funding (<$0.10 USD per capita)				−1.63% prevalence and initiation, +1.63% cessation				
**Marketing Restrictions**									
Comprehensive marketing ban		Ban on all forms of direct advertising including point of sale and indirect marketing				−5% prevalence, −8% initiation, +4% cessation		Restrictions on marketing were at minimal level from 1993 through 2009, then increased to 25% moderate and 75% minimal level in 2010 with added FDA restrictions.	
Moderate marketing ban		Ban on broadcast media, newspapers and billboards marketing and at least some indirect marketing (sponsorship, branding, giveaways)				−3% prevalence, −4% initiation, +2% cessation			
Minimal marketing ban		Ban on broadcast media advertising				−1% prevalence and −1% initiation only			
Enforcement		Government agency enforces the laws				Effects reduced 50% absent enforcement		Level 9 out of 10 for all years.	
**Cessation Treatment Policies**									
Availability of pharmaco-therapies		Legality of nicotine replacement therapy (NRT) and/or Bupropion and Varenicline		−1% prevalence, +4% cessation					Availability of NRT since 1993, and Bupropion with a prescription since 1998. Treatment coverage increased in stages from 30% coverage in 1997 to 40% in 2002, to 50% in 2007, and to 75% in 2014. A national quitline was implemented at 50% in 2003 increasing in stages to 90% in 2007. Brief interventions are set at 50% coverage for all years.
Cessation treatment financial coverage		Coverage of pharmacotherapy and behavioral cessation treatment with high publicity		−2.25% prevalence, +8% cessation					
Quit line		Three quit line types: passive, proactive and active with follow-up		−1% prevalence, +6% cessation					
Brief interventions		Advice by health care provider to quit and methods provided		−1% prevalence, +6% cessation					
All cessation policies combined		Complete availability and reimbursement of pharmaco- and behavioral treatments, quit lines, and fully implemented brief interventions		−5.68% prevalence, +29.4% cessation					
**Youth Access Policies**									
Strong enforcement and well publicized			Compliance checks conducted 4 times per year per outlet, penalties are potent and enforced with heavy publicity	−16% initiation and prevalence for ages 16–17 and −24% for ages 10–15					Low level in 1998 increasing to mid-level in 2003 and remaining at that level.
Moderate enforcement with some publicity			Compliance checks conducted regularly, penalties are potent, and publicity and merchant training are included	−8% initiation and prevalence ages 16–17 and −12% for ages 10–15					
Low enforcement			Compliance checks are conducted sporadically, penalties are weak	−2% initiation and prevalence ages 16–17 and −3% ages 10–15					

Notes: Policy effect sizes are based on previous *SimSmoke* analyses as cited in the main text. Unless otherwise indicated, the effects are in terms of the reduction in prevalence during the first year. The reduction in initiation rates and the increase in quit rates take effects during the years that the policy is in effect. Policy levels are based on information from the literature about the levels of policy as cited in the main text.

**Table 2 ijerph-18-04876-t002:** Calibration and validation of US *SimSmoke* current smoking prevalence predictions against national surveys, by age and gender, 1993–2012.

Ages	Source	1993	1998	2010	2012	Percent Change 1993–1998	Percent Change 1998–2010	Percent Change 1998–2012
**MALE**								
18+	SimSmoke	26.7%	25.2%	17.9%	17.3%	−5.8%	−28.8%	−31.2%
	TUS-CPS	26.6%	24.4%	17.2%		−8.2%	−29.6%	
	95% CI	(26.3%, 27.0%)	(23.9%, 25.0%)	(16.8%, 17.6%)			
	NHIS	27.7%	26.4%	21.5%	20.5%	−4.7%	−18.6%	−22.3%
	95% CI	(26.6%, 28.8%)	(25.5%, 27.3%)	(20.7%, 22.3%)	(19.6%, 21.4%)			
18–24	SimSmoke	27.3%	29.0%	19.8%	19.9%	6.4%	−31.6%	−31.5%
	TUS-CPS	27.7%	30.0%	19.4%		8.0%	−35.4%	
	95% CI	(26.7%, 28.7%)	(27.9%, 32.0%)	(18.0%, 20.7%)				
	NHIS	28.8%	31.3%	22.8%	20.1%	8.7%	−27.2%	−35.8%
	95% CI	(25.5%, 32.1%)	(28.4%, 34.2%)	(19.9%, 25.7%)	(17.1%, 23.1%)			
25–44	SimSmoke	30.7%	28.1%	20.2%	19.6%	−8.3%	−28.1%	−30.2%
	TUS-CPS	30.8%	28.4%	19.6%		−7.7%	−30.9%	
	95% CI	(30.3%, 31.3%)	(27.5%, 29.4%)	(19.0%, 20.3%)				
	NHIS	31.1%	29.4%	24.3%	25.4%	−5.5%	−17.3%	−13.6%
	95% CI	(29.5%, 32.7%)	(28.1%, 30.7%)	(22.8%, 25.8%)	(23.8%, 27.1%)			
45–64	SimSmoke	27.0%	26.0%	19.1%	18.2%	−3.9%	−26.3%	−29.9%
	TUS-CPS	27.1%	25.1%	18.7%		−7.3%	−25.6%	
	95% CI	(26.5%, 27.7%)	(24.1%, 26.2%)	(18.1%, 19.3%)				
	NHIS	29.2%	27.7%	23.2%	20.2%	−5.1%	−16.2%	−27.1%
	95% CI	(27.2%, 31.2%)	(26.1%, 29.3%)	(21.6%, 24.8%)	(18.8%, 21.6%)			
65+	SimSmoke	13.6%	11.6%	8.5%	8.4%	−14.7%	−26.8%	−27.6%
	TUS-CPS	13.4%	10.7%	8.6%		−20.2%	−20.1%	
	95% CI	(12.8%, 14.0%)	(9.7%, 11.7%)	(7.9%, 9.2%)				
	NHIS	13.5%	10.4%	9.7%	10.6%	−23.0%	−6.7%	1.9%
	95% CI	(11.3%, 15.7%)	(9.1%, 11.7%)	(8.3%, 11.1%)	(9.3%, 12.0%)			
**FEMALE**								
18+	SimSmoke	22.2%	20.3%	14.0%	13.5%	−8.6%	−30.8%	−33.3%
	TUS-CPS	22.3%	20.0%	13.7%		−10.5%	−31.3%	
	95% CI	(22.1%, 22.6%)	(19.5%, 20.4%)	(13.4%, 14.0%)				
	NHIS	22.5%	22.0%	17.3%	15.8%	−2.2%	−21.4%	−28.2%
	95% CI	(21.6%, 23.4%)	(21.2%, 22.8%)	(16.5%, 18.1%)	(15.1%, 16.5%)			
18–24	SimSmoke	23.7%	23.9%	15.7%	15.7%	1.1%	−34.4%	−34.4%
	TUS-CPS	23.9%	24.7%	14.7%		3.4%	−40.3%	
	95% CI	(23.0%, 24.7%)	(23.0%, 26.5%)	(13.7%, 15.8%)				
	NHIS	22.9%	24.5%	17.4%	14.5%	7.0%	−29.0%	−40.8%
	95% CI	(20.2%, 25.6%)	(21.9%, 27.1%)	(15.0%, 19.8%)	(12.3%, 16.7%)			
25–44	SimSmoke	26.3%	23.4%	16.2%	15.7%	−10.9%	−31.0%	−33.1%
	TUS-CPS	26.4%	23.8%	15.7%		−9.9%	−34.2%	
	95% CI	(26.0%, 26.8%)	(23.0%, 24.6%)	(15.1%, 16.2%)				
	NHIS	27.3%	25.6%	19.8%	17.8%	−6.2%	−22.7%	−30.5%
	95% CI	(26%, 28.6%)	(24.4%, 26.8%)	(18.4%, 21.2%)	(16.6%, 19.0%)			
45–64	SimSmoke	23.1%	21.5%	15.3%	14.4%	−6.7%	−28.8%	−33.1%
	TUS-CPS	23.2%	20.5%	15.9%		−11.9%	−22.2%	
	95% CI	(22.7%, 23.7%)	(19.6%, 21.3%)	(15.4%, 16.4%)				
	NHIS	23.0%	22.5%	19.1%	18.9%	−2.2%	−15.1%	−16.0%
	95% CI	(21.3%, 24.7%)	(21.2%, 23.8%)	(17.9%, 20.3%)	(17.6%, 20.2%)			
65+	SimSmoke	11.3%	9.8%	7.1%	7.1%	−13.5%	−27.0%	−26.9%
	TUS-CPS	11.4%	9.6%	6.8%		−16.3%	−28.6%	
	95% CI	(11.0%, 11.9%)	(8.8%, 10.3%)	(6.4%, 7.3%)				
	NHIS	10.5%	11.2%	9.3%	7.5%	6.7%	−17.0%	−33.0%
	95% CI	(9.2%, 11.8%)	(10%, 12.4%)	(8.1%, 10.5%)	(6.6%, 8.5%)			

Notes: 1. TUS-CPS = Tobacco Use Supplement of Current Population Survey that measures those who have smoked 100 cigarettes or more in their lifetime and currently smoke daily or someday; 2. NHIS = National Health Interview Survey that measures those who have smoked 100 cigarettes or more in their lifetime and now smoke every day or some days; 3. 95% CI refers to the 95% confidence interval for the prevalence.

**Table 3 ijerph-18-04876-t003:** Smoking prevalence predictions from No-NVP *SimSmoke* Counterfactual compared to national surveys, by age group and gender, 2012–2018.

Age	Source	2012	2018	Relative Reduction 2012–2018	Difference from SimSmoke *	Annual Relative Reduction †	Annual Difference from SimSmoke ^††^
**MALE**							
18+	SimSmoke	17.3%	15.2%	12.2%		2.1%	
	TUS-CPS	16.6%	12.9%	21.9%	9.7%	4.0%	1.9%
	95% CI		(12.6%, 13.3%)	(19.7%, 23.9%)	(7.5%, 11.7%)	(3.6%, 4.5%)	(1.4%, 2.3%)
	NHIS	20.5%	15.8%	22.9%	10.7%	4.2%	2.1%
	95% CI		(15.0%, 16.6%)	(19.0%, 26.8%)	(6.8%, 14.6%)	(3.5%, 5.1%)	(1.3%, 2.9%)
18–24	SimSmoke	19.9%	18.8%	5.2%		0.9%	
	TUS-CPS	16.9%	8.7%	48.4%	43.2%	10.4%	9.6%
	95% CI		(7.7%, 9.9%)	(41.6%, 54.5%)	(36.4%, 49.3%)	(8.6%, 12.3%)	(7.7%, 11.4%)
	NHIS	20.1%	8.5%	57.8%	52.6%	13.4%	12.5%
	95% CI		(6.4%, 10.5%)	(47.8%, 68.2%)	(42.6%, 63.0%)	(10.3%, 17.4%)	(9.4%, 16.5%)
25–44	SimSmoke	19.6%	18.2%	7.5%		1.3%	
	TUS-CPS	19.1%	14.5%	24.3%	16.8%	4.5%	3.3%
	95% CI		(13.9%, 15.1%)	(21.0%, 27.3%)	(13.5%, 19.8%)	(3.9%, 5.2%)	(2.6%, 3.9%)
	NHIS	25.4%	19.1%	24.8%	17.3%	4.6%	3.3%
	95% CI		(17.5%, 20.7%)	(18.5%, 31.1%)	(11.0%, 23.6%)	(3.4%, 6.0%)	(2.1%, 4.7%)
45–64	SimSmoke	18.2%	15.0%	17.4%		3.1%	
	TUS-CPS	18.2%	15.5%	14.9%	−2.5%	2.6%	−0.5%
	95% CI		(14.9%, 16.1%)	(11.6%, 18.2%)	(−5.8%, 0.8%)	(2.0%, 3.3%)	(−1.1%, 0.2%)
	NHIS	20.2%	18.3%	9.3%	−8.1%	1.6%	−1.5%
	95% CI		(16.9%, 19.7%)	(2.5%, 16.3%)	(−14.9%, −1.1%)	(0.4%, 2.9%)	(−2.7%, −0.2%)
65+	SimSmoke	8.4%	7.4%	11.4%		2.0%	
	TUS-CPS	8.6%	8.4%	3.2%	−8.2%	0.5%	−1.4%
	95% CI		(7.9%, 8.9%)	(−3.0%, 8.5%)	(−14.4%, −2.8%)	(−0.5%, 1.5%)	(−2.5%, −0.5%)
	NHIS	10.6%	9.9%	6.2%	−5.2%	1.1%	−0.9%
	95% CI		(8.7%, 11.1%)	(−4.7%, 17.9%)	(−16.1%, 6.6%)	(−0.8%, 3.2%)	(−2.8%, 1.2%)
**FEMALE**							
18+	SimSmoke	13.5%	11.8%	12.8%		2.3%	
	TUS-CPS	13.1%	10.0%	23.6%	10.7%	4.4%	2.1%
	95% CI		(9.7%, 10.2%)	(22.0%, 25.8%)	(9.1%, 13.0%)	(4.1%, 4.9%)	(1.8%, 2.6%)
	NHIS	15.8%	12.0%	24.2%	11.3%	4.5%	2.2%
	95% CI		(11.3%, 12.6%)	(20.3%, 28.5%)	(7.4%, 15.6%)	(3.7%, 5.4%)	(1.4%, 3.2%)
18–24	SimSmoke	15.7%	14.9%	5.3%		0.9%	
	TUS-CPS	12.5%	6.1%	51.2%	45.9%	11.3%	10.4%
	95% CI		(5.3%, 7.0%)	(44.1%, 57.7%)	(38.9%, 52.4%)	(9.3%, 13.4%)	(8.3%, 12.5%)
	NHIS	14.5%	7.3%	49.7%	44.4%	10.8%	9.9%
	95% CI		(5.2%, 9.4%)	(35.2%, 64.1%)	(29.9%, 58.8%)	(7.0%, 15.7%)	(6.1%, 14.8%)
25–44	SimSmoke	15.7%	14.2%	9.2%		1.6%	
	TUS-CPS	14.9%	10.6%	28.9%	19.8%	5.5%	3.9%
	95% CI		(10.2%, 11.1%)	(25.6%, 31.6%)	(16.4%, 22.4%)	(4.8%, 6.1%)	(3.2%, 4.5%)
	NHIS	17.8%	14.2%	20.2%	11.1%	3.7%	2.1%
	95% CI		(12.9%, 15.5%)	(12.9%, 27.5%)	(3.8%, 18.4%)	(2.3%, 5.2%)	(0.7%, 3.6%)
45–64	SimSmoke	14.4%	11.9%	17.6%		3.2%	
	TUS-CPS	15.4%	13.2%	14.5%	−3.1%	2.6%	−0.6%
	95% CI		(12.7%, 13.7%)	(11.1%, 17.6%)	(−6.5%, 0.0%)	(1.9%, 3.2%)	(−1.2%, 0.0%)
	NHIS	18.9%	14.3%	24.5%	6.9%	4.6%	1.4%
	95% CI		(13.1%, 15.5%)	(18.0%, 30.7%)	(0.4%, 13.1%)	(3.3%, 5.9%)	(0.1%, 2.8%)
65+	SimSmoke	7.1%	6.6%	7.7%		1.3%	
	TUS-CPS	6.8%	6.3%	6.9%	-0.8%	1.2%	−0.1%
	95% CI		(6.0%, 6.7%)	(1.5%, 11.8%)	(−6.2%, 4.1%)	(0.2%, 2.1%)	(−1.1%, 0.7%)
	NHIS	7.5%	7.3%	2.5%	−5.2%	0.4%	−0.9%
	95% CI		(6.4%, 8.2%)	(−9.3%, 14.7%)	(−17.0%, 7.0%)	(−1.5%, 2.6%)	(−2.8%, 1.3%)

Notes: 1. TUS-CPS = Tobacco Use Supplement of Current Population Survey that measures those who have smoked 100 cigarettes or more in their lifetime and currently smoke daily or someday. 2. NHIS = National Health Interview Survey that measures those who have smoked 100 cigarettes or more in their lifetime and now smoke every day or some days. 3. Due to unavailable TUS-CPS data in 2012, the prevalence in that year is estimated from a weighted average of TUS-CPS estimates in 2010/2011 (60% weight) and 2014/15 (40% weight). 4. 95% CI refers to the 95% confidence interval for the prevalence. * Difference of smoking relative reduction in the period 2012–2018 between the model and surveys; † the No-NVP projections each year were reduced in relative terms by the average annual reduction rate [(1- survey smoking relative reduction in the period 2012–2018)^1/6^ (1- *SimSmoke* smoking relative reduction in the period 2012–2018)^1/6^]; ^††^ difference of annual relative reduction between the model and surveys.

**Table 4 ijerph-18-04876-t004:** Marginal sensitivity analysis of smoking prevalence to the policy changes in the period 2012–2018 as a result of their effect sizes, ages 18 and above.

Scenario *	Range **	2012	2018	Relative Reduction in the Period 2012–2018	Difference from No Policy	2012	2018	Relative Reduction in the Period 2012–2018	Difference from No Policy
	Male Smoking prevalence					Female Smoking Prevalence			
No policy change	-	17.3%	15.7%	9.1%	-	13.5%	12.2%	9.7%	-
Price alone	0%	17.3%	15.6%	10.1%	1.0%	13.5%	12.1%	10.7%	1.0%
	−25%	17.3%	15.6%	9.8%	0.7%	13.5%	12.1%	10.4%	0.7%
	+25%	17.3%	15.5%	10.5%	1.4%	13.5%	12.0%	11.1%	1.4%
Smoke-free air laws alone	0%	17.3%	15.6%	9.9%	0.9%	13.5%	12.1%	10.6%	0.9%
	−50%	17.3%	15.7%	9.4%	0.3%	13.5%	12.2%	10.0%	0.3%
	+50%	17.3%	15.5%	10.6%	1.5%	13.5%	12.0%	11.2%	1.5%
Mass media campaigns alone	0%	17.3%	15.7%	9.2%	0.1%	13.5%	12.2%	9.8%	0.1%
	−50%	17.3%	15.7%	9.1%	0.0%	13.5%	12.2%	9.7%	0.0%
	+50%	17.3%	15.7%	9.3%	0.2%	13.5%	12.2%	9.9%	0.2%
Cessation treatment alone	0%	17.3%	15.6%	9.7%	0.6%	13.5%	12.1%	10.3%	0.6%
	−50%	17.3%	15.7%	9.4%	0.3%	13.5%	12.2%	10.0%	0.3%
	+50%	17.3%	15.6%	10.0%	1.0%	13.5%	12.1%	10.7%	1.0%
All above policies	0%	17.3%	15.2%	12.2%	3.1%	13.5%	11.8%	12.8%	3.2%
	−25%/−50%	17.3%	15.4%	10.8%	1.7%	13.5%	12.0%	11.4%	1.7%
	+25%/+50%	17.3%	15.0%	13.6%	4.5%	13.5%	11.6%	14.2%	4.6%

Notes: * “No policy change” scenario here means the policy levels will keep constant in their 2012 level in the future years; some policy change “alone” means the specific policy will change in the future years and other policies keep constant in their 2012 level; “All above policies” means all policy levels will change over years. “All above policies” with 0% change from their real levels over the period 2012–2018 was used in the main analysis to reflect the real world. ** Ranges are based on policy effect size variations from Levy et al. 2018, with a range of +/− 25% for taxes and +/− 50% for all other policies [6].

**Table 5 ijerph-18-04876-t005:** Projected smoking-attributable deaths and lives saved, unadjusted US SimSmoke compared to TUS-CPS NVP-adjusted and NHIS NVP-adjusted US SimSmoke (for ages before 45), by gender, 2012–2052.

	Adjustment	2012	2018	2052	2012–2018	2012–2052
**MALE**						
Smoking-Attributable Deaths	None	193,271	190,629	109,884	1,347,094	6,681,664
	TUS-CPS	193,271	189,629	97,150	1,343,975	6,408,032
	Range	-	(189,466–189,806)	(95,147–99,339)	(1,343,457–1,344,539)	(6,364,568–6,455,673)
	NHIS	193,271	189,604	95,862	1,343,896	6,390,611
	Range	-	(189,258–189,942)	(91,812–99,767)	(1,342,788–1,344,967)	(6,300,531–6,478,508)
Deaths Averted	TUS-CPS	-	1000	12,734	3119	273,632
	Range	-	(822–1163)	(10,545–14,737)	(2555–3637)	(225,991–317,096)
	NHIS	-	1025	14,022	3198	291,053
	Range	-	(686–1371)	(10,117–18,072)	(2127–4306)	(203,156–381,133)
**FEMALE**						
Smoking-Attributable Deaths	None	103,939	104,084	80,241	729,860	4,038,753
	TUS-CPS	103,939	103,950	72,912	729,507	3,930,434
	Range	-	(103,933–103,971)	(71,942–74,034)	(729,462–729,563)	(3,916,586–3,946,952)
	NHIS	103,939	104,004	74,777	729,649	3,964,485
	Range	-	(103,959–104,047)	(72,342–77,151)	(729,530–729,763)	(3,928,668–3,999,568)
Deaths Averted	TUS-CPS	-	134	7330	353	108,319
	Range	-	(113–151)	(6207–8299)	(298–399)	(91,801–122,168)
	NHIS	-	80	5465	211	74,269
	Range	-	(37–125)	(3091–7899)	(97–330)	(39,186–110,085)
**BOTH**						
Smoking-Attributable Deaths	None	297,211	294,713	190,126	2,076,954	10,720,417
	TUS-CPS	297,211	293,579	170,062	2,073,482	10,338,466
	Range	-	(293,400–293,777)	(167,089–173,374)	(2,072,919–2,074,101)	(10,281,154–10,402,625)
	NHIS	297,211	293,607	170,639	2,073,545	10,355,096
	Range	-	(293,217–293,989)	(164,154–176,918)	(2,072,318–2,074,730)	(10,229,199–10,478,075)
Deaths Averted	TUS-CPS	-	1134	20,064	3472	381,952
	Range	-	(935–1313)	(16,752–23,036)	(2853–4035)	(317,792–439,264)
	NHIS	-	1106	19,487	3409	365,322
	Range	-	(724–1496)	(13,208–25,972)	(2224–4636)	(242,342–491,218)

Notes: 1. TUS-CPS = Tobacco Use Supplement of Current Population Survey that measures those who have smoked 100 cigarettes or more in their lifetime and currently smoke daily or someday. 2. NHIS = National Health Interview Survey that measures those who have smoked 100 cigarettes or more in their lifetime and now smoke every day or some days. 3. No adjustment refers to SimSmoke projections without NVP adjustment in the period 2012–2018 and TUS-CPS and NHIS adjustments refer to the SimSmoke projections with NVP adjustment in the period 2012–2018 using the best estimated reduction in smoking prevalence reflected from the TUS-CPS and NHIS surveys. 4. Range refers to the implementation of TUS-CPS and NHIS adjustments using the annual relative difference in the period 2012–2018 derived from the lower and upper bound of the 95% confidence intervals from the surveys in 2018.

## Data Availability

Data and the model will be provided upon request.

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
