# Peer review of "US Nicotine Vaping Product SimSmoke Simulation Model: The Effect of Vaping and Tobacco Control Policies on Smoking Prevalence and Smoking-Attributable Deaths"

_ijerph, 2021, doi:10.3390/ijerph18094876_

Round 1

Reviewer 1 Report

Thank you for the invitation to review this manuscript which reports the results of a simulation model on the effect of vaping and tobacco control policies on smoking prevalence and smoking-attributable deaths. This is a well-written manuscript, used a well-established and validated modelling approach, and has several methodological strengths. I commend the authors for conducting this much-needed modelling study.

The authors concluded that NVP use is associated with substantial reductions in US smoking prevalence among younger adults. This is an interesting finding and backs the claim that e-cigarette diverts young adults from using smoking. However, some e-cigarette critics (and government policies) maintained the claim that they are a way into rather than a way out of smoking despite these findings and all the uncertainty analysis you’ve conducted. Although resolving this debate is clearly beyond the scope of this study, it would be helpful for readers if the authors include a few lines in the discussion regarding the most plausible e-cigarette regulatory framework that will likely result in even further reduction in smoking, and eventually e-cigarette (as I think completely stopping any nicotine use should be the end goal).

Author Response

Thank you for the invitation to review this manuscript which reports the results of a simulation model on the effect of vaping and tobacco control policies on smoking prevalence and smoking-attributable deaths. This is a well-written manuscript, used a well-established and validated modelling approach, and has several methodological strengths. I commend the authors for conducting this much-needed modelling study.

The authors concluded that NVP use is associated with substantial reductions in US smoking prevalence among younger adults. This is an interesting finding and backs the claim that e-cigarette diverts young adults from using smoking. However, some e-cigarette critics (and government policies) maintained the claim that they are a way into rather than a way out of smoking despite these findings and all the uncertainty analysis you’ve conducted. Although resolving this debate is clearly beyond the scope of this study, it would be helpful for readers if the authors include a few lines in the discussion regarding the most plausible e-cigarette regulatory framework that will likely result in even further reduction in smoking, and eventually e-cigarette (as I think completely stopping any nicotine use should be the end goal).

Response: We have added a paragraph to the Discussion section (second last paragraph) which lays out the issues required to make a complete public health assessment. We state, “Finally, our analysis only considered potential impacts of NVPs on smoking. A complete public health assessment would require consideration of the impact of NVP use on health outcomes [30]. If NVPs increase mortality risk, then the increased deaths due to NVP use would reduce the number of total deaths averted. NVP use would improve public health to the extent that they replace smoking by those who would have otherwise initiated cigarette use or who would not have otherwise quit smoking, and reduce public health to the extent that it acts as a gateway to smoking or is used as a substitute for smoking cessation. In particular, policies may be needed to reduce NVP use among youth if NVPs act as a gateway to smoking.”

Reviewer 2 Report

This manuscript uses the simulation model SimSmoke – which has been previously developed to examine country-level smoking trends and tobacco policy effects – to indirectly estimate the effect of nicotine vaping product (NVP) use on cigarette smoking prevalence and smoking-attributable deaths in the US. After validating the model to US smoking prevalence trends and accounting for several types of cigarette policies, the effect of NVP use is estimated as the difference between a counterfactual trend (which continues smoking prevalence trends calibrated pre-NVP (~2012) and projects them forward in time (through 2018), and actual smoking prevalence trends (as measured by the TUS-CPS and the NHIS). The study shows that NVP use resulted in approximately a 10% decline in smoking rates, compared to the counterfactual scenario (NVPs had never appeared), and resulted in nearly 0.4 million smoking-attributable deaths being averted by 2052 (after 40 years of NVP availability). The modeling work is extremely strong and the findings are very important to the scientific debate on e-cigarettes.

Major comments:

  • The Methods section alludes to a Supplementary Report but I was not able to find a file other than the man manuscript.
  • More detail/clarity of the SimSmoke model and its parameters would help, for example:
    • Since there are several other publications using SimSmoke, it would be helpful to clarify what has been reported previously vs. what is new to this manuscript. E.g. is the validation to TUS-CPS and NHIS smoking prevalence trends unique to this study, or has this already been reported elsewhere?
    • Table 1 is very helpful, but it is not clear to me whether the numbers in the “Policy Effect Size” and “Policy Level” columns are derived from empirical estimates from other work, or were arrived at via model validation/calibration (e.g. price elasticities, worksite ban effect size and policy level).
    • Table 1, the rows describing “enforcement” are confusing – on one hand, the “policy effect size” column seems to describe a binary effect (“effects reduced 50% absent publicity and enforcement”) yet the “policy level” column seems to describe a continuous effect (“level 8 out of 10 in all years”). Can the authors clarify how these values work together in the model?
    • Table 1 under the super-row “Cessation Treatment Policies,” it’s not clear from the “policy level” description whether the “all cessation policies combined” ever occurs during the simulation timeframe (as brief interventions are set to 50% coverage; does this count?)
    • Are the values in Table 1 established from prior publications of SimSmoke, or are they unique to this study?
    • Several aspects of the model description raise the question whether the model is deaggregated at the state level (or even smaller), e.g. “Data on smoke-free air laws is from the American Nonsmokers Rights Foundation website by state and locality…”, “strict bans in 2019 were implemented in 76.1% of worksites”, and “the policy was set to 90% minimal and 10% moderate” (what is the denominator here?)
  • The overall simulation results for smoking prevalence are much closer to the TUS-CPS than they are to NHIS. Why is this the case, and does this present a challenge to the overall conclusions of the paper?
    • Related to above, Section 3.1 states “the levels of smoking prevalence in 2010 and the relative reductions between 1998 and 2010 from SimSmoke were generally between those of the NHIS and TUS-CPS in all age groups” is potentially misleading because it does not mention that SimSmoke values are much closer to TUS-CPS, giving the impression that SimSmoke is between the two national survey values.
    • The text states “All SimSmoke estimates of 2010 smoking prevalence were within the 95% CIs for TUS-CPS estimates, except for… [particular age/sex groups]” – is the omission of an equivalent statement about NHIS indicative of the 2010 SimSmoke estimates falling outside 95% CI’s of NHIS estimates?
  • Table 4 findings: why does making policies worse (e.g. “all above policies -25%/-50%) result in a larger relative reduction in smoking prevalence? It seems counterintuitive that in this table, the lowest reduction in relative reduction is under the base case (no policy change).
  • The main analyses of averted smoking-related deaths appropriately focus on younger age groups (the only ones whose relative reductions in smoking prevalence exceeded the 95% CI of TUS-CPS and/or NHIS estimates). However, the disease burden is much lower among younger smokers. Is this accounted for in the analyses? I see that the analyses are age-weighted but it’s not clear to me whether the averted deaths only include people who would have died by 2052, or whether this number could potentially be higher if following the same people (those who switch to NVPs) over a longer time period.
  • Last paragraph of results: in the sensitivity analysis for averted deaths, why did the averted deaths decrease when including more people in the calculation (i.e. all age groups)?

Minor comments:

  • It may be appropriate to cite other similar literature such as Selya & Foxon 2021 Addiction, which also used simulation to examine actual and counterfactual trends in smoking prevalence among youth, and shows that NVPs are likely to have resulted in lower-than-expected smoking trends.
  • Section 2.1, 2nd paragraph: “… and validated the SimSmoke population estimates against Census Bureau estimates through 2052.” How is validation done through 2052 when data aren’t available yet? Please clarify.
  • Section 2.1: repeated sentence: “transitions are measured net of death rates distinguished by smoking status”
  • Table 3 column headings could be clarified, especially to name the two columns “Difference from SimSmoke” differently (e.g. “Relative Reduction Difference from SimSmoke” and “Annual Reduction Difference from SimSmoke”)
  • The unavailable TUS-CPS data in 2012 is addressed only in the footnote to Table 3 (unless I missed it) but should also be described in the main text.

Author Response

//

This manuscript uses the simulation model SimSmoke – which has been previously developed to examine country-level smoking trends and tobacco policy effects – to indirectly estimate the effect of nicotine vaping product (NVP) use on cigarette smoking prevalence and smoking-attributable deaths in the US. After validating the model to US smoking prevalence trends and accounting for several types of cigarette policies, the effect of NVP use is estimated as the difference between a counterfactual trend (which continues smoking prevalence trends calibrated pre-NVP (~2012) and projects them forward in time (through 2018), and actual smoking prevalence trends (as measured by the TUS-CPS and the NHIS). The study shows that NVP use resulted in approximately a 10% decline in smoking rates, compared to the counterfactual scenario (NVPs had never appeared), and resulted in nearly 0.4 million smoking-attributable deaths being averted by 2052 (after 40 years of NVP availability). The modeling work is extremely strong and the findings are very important to the scientific debate on e-cigarettes.

Major comments:

  • The Methods section alludes to a Supplementary Report but I was not able to find a file other than the main manuscript.

Response: We did not submit the Supplementary Report with the paper. We have changed the statement in the Methods statement to state that the Technical Report is available from the authors.

More detail/clarity of the SimSmoke model and its parameters would help, for example:

  • Since there are several other publications using SimSmoke, it would be helpful to clarify what has been reported previously vs. what is new to this manuscript. E.g., is the validation to TUS-CPS and NHIS smoking prevalence trends unique to this study, or has this already been reported elsewhere?

Response:. In other publications, SimSmoke was previously validated against the CPS-TUS for various time periods. In this paper, the model is validated relative to recent CPS-TUS and the NHIS trends in order to extrapolate into the future. However, the validation is not the focus of this paper, but just used so that the model can be applied to determine NVP effects in the post-NVP period (after 2012). Rather than provide detail on the different validation periods, we simply indicate that the US SimSmoke is a “previously validated” model, which we now state explicitly in the third sentence of the last paragraph of the Introduction.

  • Table 1 is very helpful, but it is not clear to me whether the numbers in the “Policy Effect Size” and “Policy Level” columns are derived from empirical estimates from other work, or were arrived at via model validation/calibration (e.g. price elasticities, worksite ban effect size and policy level).

Response: The policy effect sizes were derived from other work cited in the text and the policy levels are based on information from various sources also cited in the text. We have revised the Notes section at the bottom of the Table to state: “Policy effect sizes are based on previous SimSmoke analyses as cited in the text. Unless otherwise indicated, the effects are in terms of the reduction in prevalence during the first year. The reduction in initiation rates and the increase in quit rates take effects during the years that the policy is in effect. Policy levels are based on information about the levels of policy from the literature as cited in the text.”

  • Table 1, the rows describing “enforcement” are confusing – on one hand, the “policy effect size” column seems to describe a binary effect (“effects reduced 50% absent publicity and enforcement”) yet the “policy level” column seems to describe a continuous effect (“level 8 out of 10 in all years”). Can the authors clarify how these values work together in the model?

Response: The effect sizes are deflated to incorporate the lack of enforcement and publicity using the following multiplicative function: Effect size x (1- 0.5*Publicity Indicator + 0.5* Enforcement Level), where the enforcement level is converted to percentage terms. We have clarified in Table 1.

  • Table 1 under the super-row “Cessation Treatment Policies,” it’s not clear from the “policy level” description whether the “all cessation policies combined” ever occurs during the simulation timeframe (as brief interventions are set to 50% coverage; does this count?)

Response: The “all cessation treatment policies combined” is with all components set to their maximum level including brief interventions. We added that brief interventions are “fully implemented” in the policy description for all cessation policies implemented in Table 1.

  • Are the values in Table 1 established from prior publications of SimSmoke, or are they unique to this study?

Response: They are all established from prior publications as cited in the text. See our response to Major comment 2 above.

  • Several aspects of the model description raise the question of whether the model is deaggregated at the state level (or even smaller), e.g. “Data on smoke-free air laws is from the American Nonsmokers Rights Foundation website by state and locality…”, “strict bans in 2019 were implemented in 76.1% of worksites”, and “the policy was set to 90% minimal and 10% moderate” (what is the denominator here?)

Response: The data are de-aggregated for smoke-free air laws (only), and are obtained by multiplying the population for each state or local area by an indicator of law and then summed over states and divided by the US population. We have added to the text that the percent is of states and localities in the US.

  • The overall simulation results for smoking prevalence are much closer to the TUS-CPS than they are to NHIS. Why is this the case, and does this present a challenge to the overall conclusions of the paper?

Response: The model uses TUS-CPS data to develop the smoking prevalence, initiation and cessation measures, which is why it corresponds more closely with the TUS-CPS. The NHIS data and indeed many sources of data differ not only in terms of levels but trends. Nonetheless, the estimates from the two surveys are generally consistent and reasonably close to each other. The fact that the model estimates are closer to the TUS-CPS does not change our study conclusions.

  • Related to above, Section 3.1 states “the levels of smoking prevalence in 2010 and the relative reductions between 1998 and 2010 from SimSmokewere generally between those of the NHIS and TUS-CPS in all age groups” is potentially misleading because it does not mention that SimSmoke values are much closer to TUS-CPS, giving the impression that SimSmoke is between the two national survey values.

Response: We have added in Results section 3.1 that the estimates are “… closer to those of the TUS-CPS for the different age groups.”

  • The text states “All SimSmoke estimates of 2010 smoking prevalence were within the 95% CIs for TUS-CPS estimates, except for… [particular age/sex groups]” – is the omission of an equivalent statement about NHIS indicative of the 2010 SimSmoke estimates falling outside 95% CI’s of NHIS estimates?

Response: The reviewer makes a good point. We have added, “There was less alignment with NHIS estimates [compared to the TUS-CPS].”

  • Table 4 findings: why does making policies worse(e.g. “all above policies -25%/-50%) result in a larger relative reduction in smoking prevalence? It seems counterintuitive that in this table, the lowest reduction in relative reduction is under the base case (no policy change).

Response: Thanks for the comment. Compared to the base case, smaller policy effects (-25%/-50%) lead to smaller reductions in smoking prevalence than the base case due to the reduced effect of policy changes, whereas larger policy effects (+25%/+50%) lead to greater reductions in smoking prevalence than the base case due to the increased effect of policy changes. Therefore, with no policies implemented, there is the least reduction in smoking prevalence and the least difference between the actual smoking prevalence and the model predictions. Similarly, with all policies implemented and policies at their greatest impact (+25%, +50%), the model predicts the greatest reduction in smoking prevalence and thus the greatest difference between actual smoking prevalence and the model predictions. To clarify, we added the following second sentence to the first paragraph in section 3.3: “The relative reduction in smoking prevalence from 2012-2018 increases with the magnitude of policy impact, and consequently the total NVP impact, i.e., their difference from actual smoking prevalence, increases with the magnitude of policy impact.”

  • The main analyses of averted smoking-related deaths appropriately focus on younger age groups (the only ones whose relative reductions in smoking prevalence exceeded the 95% CI of TUS-CPS and/or NHIS estimates). However, the disease burden is much lower among younger smokers. Is this accounted for in the analyses? I see that the analyses are age-weighted but it’s not clear to me whether the averted deaths only include people who would have died by 2052, or whether this number could potentially be higher if following the same people (those who switch to NVPs) over a longer time period.

Response: The reviewer makes a good point that much of the effect on smoking-attributable deaths occur at later ages (which is taken into account in our analyses), and thus our estimates tend to be conservative. We have added the following sentence in the third paragraph in the Discussion, “We note that these reductions increase over time, because much of the effect of NVPs is for those at younger ages and deaths attributable to smoking occur at later ages.”

  • Last paragraph of results: in the sensitivity analysis for averted deaths, why did the averted deaths decrease when including more people in the calculation (i.e. all age groups)?

Response: The number of deaths averted decreased when including those at older ages, because we obtain negative rather than positive impacts of NVPs for many of the older age groups. We now explain, “As a result of findings that NVP use may increase smoking for some older age groups, ….”

Minor comments:

  • It may be appropriate to cite other similar literature such as Selya & Foxon 2021 Addiction, which also used simulation to examine actual and counterfactual trends in smoking prevalence among youth, and shows that NVPs are likely to have resulted in lower-than-expected smoking trends.

Response: We thank the reviewer for this cite. The cite has been added.

  • Section 2.1, 2ndparagraph: “… and validated the SimSmoke population estimates against Census Bureau estimates through 2052.” How is validation done through 2052 when data aren’t available yet? Please clarify.

  • Section 2.1: repeated sentence: “transitions are measured net of death rates distinguished by smoking status”

Response: Thanks for catching this. The first appearance of the sentence has been removed.

  • Table 3 column headings could be clarified, especially to name the two columns “Difference from SimSmoke” differently (e.g. “” and “Annual Reduction Difference from SimSmoke”)

Response:  We changed “Annual reduction” to “Annual relative reduction” and Difference from SimSmoke to “Annual Difference from SimSmoke“ to be more technically correct. We also added an explanation of the labels in footnotes to the table.

  • The unavailable TUS-CPS data in 2012 is addressed only in the footnote to Table 3 (unless I missed it) but should also be described in the main text.

Response: We have added as the third sentence in section 2.3 of the text that “the prevalence in that year is estimated as a weighted average of TUS-CPS estimates in 2010/11 (60% weight) and 2014/15 (40% weight).”